# LET YOUR HEART SPEAK IN ITS MOTHER TONGUE: MULTILINGUAL CAPTIONING OF CARDIAC SIGNALS

## ABSTRACT

Cardiac signals convey a significant amount of information about the health status of a patient. Upon recording these signals, cardiologists are expected to manually generate an accompanying report to share with physicians and patients. Generating these reports, however, can be time-consuming and error-prone, while also exhibiting a high degree of intra- and inter-physician variability. To address this, we design a neural, multilingual, cardiac signal captioning framework. In the process, we propose a discriminative multilingual representation learning method, RTLP, which randomly replaces tokens with those from a different language and tasks a network with identifying the language of all tokens. We show that RTLP-generated reports are of high quality and clinical utility, and are on par with reports generated by networks pre-trained with state-of-the-art methods such as MLM and MARGE. We also show that generated reports exhibit higher quality and clinical utility when RTLP is fine-tuned in a multilingual setting than in a monolingual setting, a phenomenon we refer to as the *blessing of multilinguality*.

## 1 INTRODUCTION

Cardiac signals convey a significant amount of information about the status of a patient. Upon recording these signals, an expert cardiologist is often required to interpret the findings, manually generate an accompanying textual report (Richley & Walters, 2020), and share it with fellow physicians (and patients). Such a report is crucial within medicine, impacting communication between physicians, facilitating clinical decision-making, and holding care teams accountable for their actions (Waegemann et al., 2002). Manually generating these reports, however, can be time-consuming and error-prone, while also exhibiting a high degree of intra and inter-physician variability. Combined, these can detract from a physician's time with a patient, thus decreasing patient satisfaction, and hinder communication between physicians (Hibbard et al., 2001; Keselman & Smith, 2012), thus potentially compromising patient outcomes (Brailer et al., 1997).

One way to address this challenge is by collapsing a clinical report into multiple pertinent medical conditions and designing a system that identifies such conditions. However, abstracting away the details of a report can obscure the full assessment of a patient's health status and defining the pertinence of conditions can be disease-specific and non-trivial. Based on these observations, we look to address the following question: *how do we automatically generate clinically accurate reports that reliably summarize cardiac signals?* To address this question, the fields of visual-language representation learning and captioning hold promise. In this setting, rich representations of visual inputs and language are learned to automatically generate captions. Representations of the latter can be learned in a discriminative monolingual setting, as with ELECTRA (Clark et al., 2020), or in a generative multilingual setting, as with MARGE (Lewis et al., 2020). However, a discriminative multilingual framework has yet to be explored. Moreover, previous work has focused exclusively on the captioning of medical images (Hasan et al., 2018; Zeng et al., 2020). It has not explored the captioning of cardiac time-series signals.

In this paper, we address the outlined question by designing a neural, multilingual, cardiac signal captioning system while exploiting a large-scale electrocardiogram (ECG) database. Our contributions are the following: (1) We propose a discriminative multilingual representation learning method, replaced token language prediction (RTLP), in which we randomly replace tokens with those from other languages, and task a network with predicting the language of all tokens. (2) We show that

RTLP allows networks to generate reports, in multiple languages, that are as clinically accurate as those generated by masked language modelling (MLM). To the best of our knowledge, we are the first to propose a multilingual cardiac signal captioning system. (3) We conduct extensive studies to show that generated reports exhibit higher quality and clinical utility when RTLP is fine-tuned in a multilingual setting than in a monolingual setting, a phenomenon we refer to as the *blessing of multilinguality*.

## 2 RELATED WORK

**Language representation learning**   Representation learning is integral to modern natural language processing. Generative tasks such as MLM show promise (Devlin et al., 2019; Liu et al., 2019b), even in the multilingual setting (Conneau & Lample, 2019; Conneau et al., 2020b; Liu et al., 2020). MARGE (Lewis et al., 2020) retrieves documents, in potentially different languages, and attempts to reconstruct a target document. Others propose to jointly learn textual and visual representations (Sun et al., 2019; Lu et al., 2019; Zhang et al., 2020b). Similar to our work is ELECTRA (Clark et al., 2020), where tokens are replaced with those from a generative model, and a network is tasked with discriminating between the original and replaced tokens. This approach remains computationally expensive where both an MLM and discriminative network are learned. It also does not optimize a multilingual objective. To the best of our knowledge, we are the first to propose a discriminative multilingual representation learning method in the context of cardiac signals.

**Multilingual representation learning**   Pre-training and fine-tuning networks on multiple languages confers benefits to NLP tasks (Conneau et al., 2020b; Pratap et al., 2020; Conneau et al., 2020a; Artetxe et al., 2020). For example, Conneau et al. (2020b) and Artetxe et al. (2020) show that multilingual pre-training outperforms its monolingual counterpart in solving downstream NLP tasks. These findings have been partially driven by monolingual datasets that are machine-translated to multiple languages, as with XNLI (Conneau et al., 2018). We similarly translate ground-truth ECG reports into multiple languages in order to guide the generation of multilingual reports. Although Huang et al. (2019) propose a pre-training setup similar to ours, we explore more languages and define a different pre-training task (RTLP) which we exploit for cardiac signal captioning.

**Captioning in healthcare**   Biomedical image captioning has traditionally focused on chest X-rays (Kisilev et al., 2016; Jing et al., 2017; Hasan et al., 2018; Jing et al., 2020; Zeng et al., 2020; Liu et al., 2021). Liu et al. (2019a) condition their captioning system on the medical topic to be discussed and Wang et al. (2018) propose a multi-level attention model that attends to both the image and the text. Previous work which captioned electroencephalogram (EEG) signals (Biswal et al., 2019; 2020) did not explore a multilingual representation learning method and does not extend to the generation of reports in *multiple* languages. To the best of our knowledge, we are the first to propose the multilingual captioning of cardiac signals.

## 3 BACKGROUND

### 3.1 CARDIAC SIGNAL CAPTIONING

We begin by assuming access to a dataset, $\mathcal{D} = \{\boldsymbol{x}_i, \text{cap}_i\}_{i=1}^N$ comprising $N$ cardiac signals, $\boldsymbol{x} \in \mathbb{R}^D$, and their associated captions, $\text{cap} = \{w_s\}_{s=1}^S$, which consist of $S$ words, $w_s$. The goal of cardiac signal captioning is to generate a caption (report) that reliably summarizes the physiological state of a patient as manifested in a cardiac signal. To extract features from the signal, an encoder, $f_\theta : \boldsymbol{x} \in \mathbb{R}^D \to \{\boldsymbol{v}_t \in \mathbb{R}^M\}_{t=1}^T$, parameterized by $\boldsymbol{\theta}$, maps a $D$-dimensional instance, $\boldsymbol{x}$, to a set of $T$ representations, $\{\boldsymbol{v}_t \in \mathbb{R}^M\}_{t=1}^T$, each of which is $M$-dimensional (see Fig. 1 left).

To convert the captions into a format ingestible by a network, we first convert each word, $w_s$, in a caption, $\text{cap} = \{w_s\}_{s=1}^S$, to a token, $u_s$, to form a sequence of tokens $\{u_s\}_{s=1}^S$. Such tokenization can involve lower-casing and stemming words, in addition to removing punctuation. After deriving tokens from all captions in a training set, we form a fixed vocabulary, $V = \{u_i\}_{i=1}^C$, of the $C$ unique tokens where $|V| = C$. We then define an embedding matrix (lookup table), $\boldsymbol{E} \in \mathbb{R}^{C \times M} : u \to \boldsymbol{e} \in \mathbb{R}^M$, which maps each token, $u$, to an $M$-dimensional token embedding, $\boldsymbol{e}$. These embeddings are

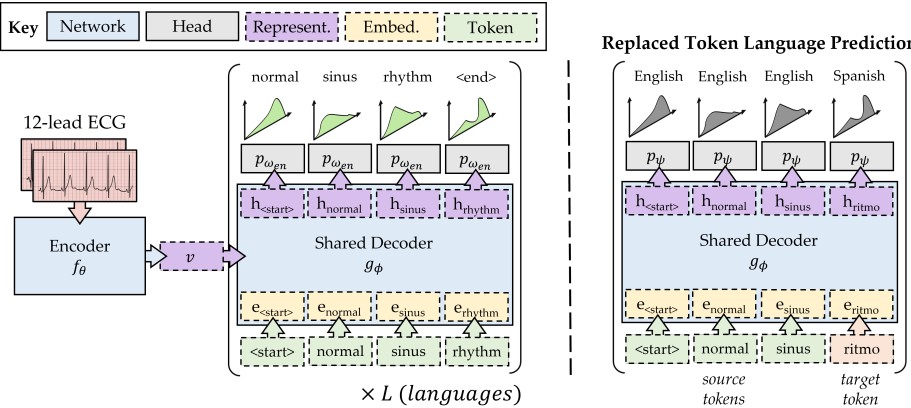

Figure 1: **(Left) Multilingual cardiac signal captioning pipeline.** We feed a 12-lead ECG into an encoder, $f_\theta$, to extract representations, $\boldsymbol{v}$. These are fed, alongside embeddings, $\boldsymbol{e}$, of tokens from a particular language, to a decoder, $g_\phi$, to generate token representations, $\boldsymbol{h}$. We feed $\boldsymbol{h}$ into a language-specific head, $p_{\boldsymbol{\omega}_l}$, to generate a caption in a specific language. **(Right) Replaced token language prediction framework.** We randomly replace source tokens with those from a target language and task the network with classifying the language of all tokens. In doing so, we encourage the network to capture relationships between representations of tokens from different languages.

typically randomly-initialized and learned in an end-to-end manner via gradient descent. As such, every caption can now be represented as a sequence of token embeddings, $\{\boldsymbol{e}_s\}_{s=1}^S$.

To extract features from language, a decoder, $g_\phi : \{\boldsymbol{v}_t\}_{t=1}^T, \{\boldsymbol{e}_s\}_{s=1}^S \rightarrow \{\boldsymbol{h}_s \in \mathbb{R}^M\}_{s=1}^S$, parameterized by $\phi$, attends to all *encoder* representations, $\{\boldsymbol{v}_t\}_{t=1}^T$, while mapping a token embedding, $\boldsymbol{e}_s$, at step, $s$, to an $M$-dimensional token representation, $\boldsymbol{h}_s$. Each token representation, $\boldsymbol{h}_s$, in the sequence of representations, $\{\boldsymbol{h}_s\}_{s=1}^S$, is then fed into a linear classification head, $p_{\boldsymbol{\omega}} : \boldsymbol{h}_s \in \mathbb{R}^M \rightarrow \boldsymbol{y}_s \in \mathbb{R}^C$, to output a probability distribution, $\boldsymbol{y}_s$, over the $C$ tokens in the vocabulary. This forms a sequence of outputs, $\{\boldsymbol{y}_s\}_{s=1}^S$. At each step, $s$, in the sequence, the goal is to maximize the likelihood of generating the token of the next step, $u_{s+1}$. Therefore, by identifying the most probable output token at each step, we can form a sentence of words (i.e., a caption).

## 4 METHODS

### 4.1 MULTILINGUAL CAPTIONING OF CARDIAC SIGNALS

At the surface, a monolingual captioning framework would appear to be sufficient for physicians communicating in a single language. However, the motivation for a *multilingual* captioning framework, in which reports are simultaneously generated in multiple languages, is threefold. First, a multilingual framework would obviate the cumbersome process of having to train a distinct model for different languages (Conneau et al., 2020a). Second, recent work has demonstrated the benefits of incorporating additional languages into the learning process (Artetxe et al., 2020). Lastly, a multilingual framework exhibits greater flexibility than its monolingual counterpart, as it can always be collapsed, during inference, to generate captions from a single language.

To enable multilingual captioning, we first assume access to $L$ language-specific datasets, $\{\mathcal{D}_l\}_l^L$ where $\mathcal{D}_l = \{\boldsymbol{x}_i, \text{cap}_i^l\}_{i=1}^N$ comprises $N$ cardiac signals, $\boldsymbol{x}$, and captions in a specific language, $l \in \mathbb{L} = \{\text{en}, \text{es}, \cdots\}$, where en and es represent English and Spanish, respectively. Note that the cardiac signals are *shared* across the datasets. We follow the same encoder-decoder approach mentioned in the previous section with one exception. We replace the single classification head with $L$ language-specific heads to account for the distinct vocabularies of the $L$ languages. In doing so, we exploit recent observations that demonstrated the utility of having a network with parameters that are both language-specific and shared across languages (Zhang et al., 2020a).

Formally, for each language, $l$, we have a linear classification head, $p_{\boldsymbol{\omega}_l} : \boldsymbol{h}_s^l \in \mathbb{R}^M \to \boldsymbol{y}_s^l \in \mathbb{R}^C$, with $\boldsymbol{\omega}_l \in \{\boldsymbol{\omega}_{\text{en}}, \boldsymbol{\omega}_{\text{es}}, \cdots\}$ reflecting language-specific parameters. Each head maps the token representation, $\boldsymbol{h}_s^l$, at each step, $s$, in the sequence to a probability distribution, $\boldsymbol{y}_s^l$, over $C$ tokens where $C \in \{|V_{\text{en}}|, |V_{\text{es}}|, \cdots\}$ reflects the size of a language-specific token vocabulary. When doing this for each step in the sequence, we arrive at a set of probability distributions, $\{\boldsymbol{y}_s^l\}_{s=1}^S$. As with traditional language models, at each step in the sequence, $s \in [1, S]$, we maximize the likelihood of observing the next token in the sequence, $u_{s+1}^l$, from a particular language, $l \in \mathbb{L}$. Therefore, for a mini-batch of $B$ captions in a single language, we would optimize the categorical cross-entropy loss at each step, $s$. To extend this to $L$ languages, we load $L$ mini-batches and optimize the following *multi-task* categorical cross-entropy loss.

$$\mathcal{L}_{\text{multilingual}} = -\frac{1}{LBS} \sum_{l \in \mathbb{L}}^{L} \sum_{i=1}^{B} \sum_{s=1}^{S} \log p_{\boldsymbol{\omega}_l}(y_{i,s}^l = u_{i,s+1}^l) \tag{1}$$

## 4.2 Replaced Token Language Prediction

To facilitate achieving the downstream task of multilingual cardiac signal captioning, we design a discriminative multilingual pre-training task that learns the decoder parameters, $\phi$, and the token embeddings, $\boldsymbol{e}$. At a high-level, this task involves randomly selecting tokens in a sequence, replacing them with semantically-similar tokens from a different language, and tasking a network with classifying the language of all tokens (see Fig. 1 right).

The intuition behind our framework is that a network exposed to semantically-similar tokens from distinct languages which share the same context (neighbouring tokens) can learn that such tokens are indeed similar to one another. We hypothesize that encouraging this behaviour can lead to the learning of token representations that are beneficial for the downstream task of *multilingual* captioning since these tokens (from different languages) are likely to arise in the same generated report, albeit in a different language. As such, the generated multilingual reports might be more likely to contain plausible tokens, and thus be clinically accurate. From hereon forward, we refer to this method as replaced token language prediction (RTLP) and describe its mechanics next.

**Source token selection** Given tokens, $\{u_s^{l_{src}}\}_{s=1}^S$, in a sequence of length, $S$, from a source language, $l_{src} \in \mathbb{L}$, we first sample $K$ distinct steps, $\{s_k\}_{k=1}^K$ where $s_k \in [1, S]$ from a uniform distribution, $\mathcal{U}$. We then replace the corresponding source tokens, $\{u_{s_k}^{l_{src}}\}_{k=1}^K$, with those from a target language, $\{u_{s_k}^{l_{tgt}}\}_{k=1}^K$. We outline how to select the target language and token next.

**Target language and token selection** For each source token, $u_{s_k}^{l_{src}}$, we sample a target language, $l_{tgt} \sim \mathcal{U}(\mathbb{L}')$, uniformly at random from the set of remaining languages $\mathbb{L}' = \mathbb{L} \setminus l_{src}$ where $|\mathbb{L}'| = L - 1$. Given $l_{tgt}$, we now sample a target token, $u_{s_k}^{l_{tgt}} \sim \mathcal{U}(V_{l_{tgt}})$, uniformly at random from the language-specific vocabulary of tokens, $V_{l_{tgt}}$. Such random sampling, however, can lead to the selection of a target token that is semantically different from the source token. As such, the network may discriminate between source and target tokens by using a detrimental shortcut that is based on semantics instead of language.

To avoid this behaviour, we instead adopt a strategy where the target token is likely to be semantically similar to (e.g., a noisy translation of) the source token. Formally, we quantify the cosine similarity, $\text{sim}_j$, between the source token embedding, $\boldsymbol{e}_{s_k}^{l_{src}} \in \mathbb{R}^M$, and the embedding, $\boldsymbol{e}_j^{l_{tgt}} \in \mathbb{R}^M$ of each token, $u_j^{l_{tgt}} \in V_{l_{tgt}}$ in the vocabulary of the target language. By taking the softmax of these similarities, we form a categorical distribution, $q$, with elements, $q_j$, from which we sample $u_{s_k}^{l_{tgt}}$. As the token embeddings become more meaningful during training, the sampled target token is more likely to be semantically similar to the source token.

$$u_{s_k}^{l_{tgt}} \sim q \quad , \quad \text{where} \quad q_j = \frac{\exp^{(\text{sim}_j)}}{\sum_m^{|V_{l_{tgt}}|} \exp^{(\text{sim}_m)}} \quad , \quad \text{sim}_j = \frac{\boldsymbol{e}_{s_k}^{l_{src}} \cdot \boldsymbol{e}_j^{l_{tgt}}}{|\boldsymbol{e}_{s_k}^{l_{src}}||\boldsymbol{e}_j^{l_{tgt}}|} \tag{2}$$

**Objective function** Equipped with a sequence of tokens in source and target languages, we predict the language of such tokens. To do so, we define a classification head, $p_{\boldsymbol{\psi}} : \boldsymbol{h} \to \boldsymbol{y} \in \mathbb{R}^L$,

parameterized by $\psi$, that maps each token representation, $\boldsymbol{h}$, to a probability distribution, $\boldsymbol{y}$, over $L$ languages. Formally, given a mini-batch of $B$ captions of length, $S$, in the source language, $l$, we optimize the categorical cross-entropy loss. This is achieved for the tokens in the source $(r = 0)$ and target $(r = 1)$ languages. To extend this to $L$ source languages, we load $L$ mini-batches and optimize the following *multi-task* categorical cross-entropy loss where $\mathbb{1}$ is the indicator function.

$$\mathcal{L}_{\text{RTLP}} = -\frac{1}{LBS} \sum_{l \in \mathbb{L}} \sum_{i=1}^{B} \sum_{s=1}^{S} \mathbb{1}_{r=0} \cdot \log p_{\psi}(y_{i,s} = l_{src}|u_{i,s}^{l_{src}}) + \mathbb{1}_{r=1} \cdot \log p_{\psi}(y_{i,s} = l_{tgt}|u_{i,s}^{l_{tgt}}) \quad (3)$$

## 5 Experimental Design

**Datasets** We evaluate our framework on the PTB-XL dataset (Wagner et al., 2020), the *only* publicly-available dataset which comprises ECG signals with a corresponding clinical report. Each ECG signal (from a total of 18,885 patients) is also associated with cardiac abnormality labels which we group into 5 classes (Strodthoff et al., 2020). We provide further details in Appendix A.1.

**Representation learning of cardiac signals** Supervised pre-training remains an effective way to learn rich, generalizable representations. As such, we pre-train the *encoder*, $f_{\theta}$, to map 12-lead ECG signals in the training set to cardiac abnormalities. We decide on this task because ECG reports are likely to reflect such abnormalities. Therefore, grounding the captioning process in representations of cardiac signals that can discriminate between cardiac abnormalities can be beneficial.

**Representation learning of clinical reports** To tokenize the ECG reports, we exploit $L$ language-specific tokenizers from SpaCy (see Appendix C.5 for details). For simplicity, we lower-case the text and remove any punctuation. By keeping track of unique tokens, we form $L$ distinct vocabularies. Each vocabulary also includes language-specific tokens to indicate the start and end of the report, and the [PAD] and [OOV] tokens to refer to padded entries and tokens observed during inference that are not seen during training, respectively. We also introduce the [MASK] token where appropriate. The details of network architectures can be found in Appendix C.1.

**Multilingual cardiac signal captioning** After pre-training the encoder, $f_{\theta}$, and the decoder, $g_{\phi}$, independently of one another, we exploit the learned parameters, $\{\theta, \phi\}$, and token embeddings to solve the task of cardiac signal captioning (see Fig. 1 left). After experimenting with several variants of our framework, we chose to *freeze* the encoder parameters and extract multiple representations per cardiac signal. Multilingual captioning also requires multilingual ground-truth ECG reports. Since such *paired* reports do not exist, we translate the original set of reports in English (en) to six languages {German (de), Greek (el), Spanish (es), French (fr), Italian (it), Portuguese (pt)} using the Google Translate API, a strategy similar to that adopted by Conneau et al. (2018). We open-source these reports[1] and provide further details in Appendix B. Although translated reports can be imperfect ground-truth reports, we hypothesize (and indeed show) that the net effect of multilinguality on the quality and clinical utility of generated reports is advantageous.

**Baseline Methods** As the first to propose multilingual captioning of cardiac signals, we cannot trivially compare to previous methods. However, we compare our pre-training method, RTLP, to the following state-of-the-art pre-training methods: 1) **MLM** (Devlin et al., 2019), masked language modelling where the decoder is tasked with identifying masked tokens, 2) **ELECTRA** (Clark et al., 2020), where the decoder is tasked with identifying whether tokens have been replaced with those from an MLM model, and 3) **MARGE** (Lewis et al., 2020), a multilingual generative language representation learning approach where source documents in various languages are retrieved to generate a similar target document (see Appendix C.3 for details on how we adapted these methods).

**Evaluation Metrics** We quantify the quality of the generated reports by comparing the degree of overlap of tokens in such reports to those in the ground-truth report. Specifically, we use the BLEU$-1$ (Papineni et al., 2002), METEOR (Banerjee & Lavie, 2005), and ROUGE$-$L (Lin, 2004) scores (see Appendix C.4 for more details).

---

[1]Code and data: `https://tinyurl.com/CardiacSignalCaptioning`

**Hyperparameters** We conduct our experiments using PyTorch (Paszke et al., 2019). We pre-train the encoder and decoder with a patience value of 10 and 25 epochs, respectively, on the validation loss. When transferring the parameters to the task of cardiac signal captioning, we use those associated with the lowest validation loss. When fine-tuning, we checkpoint the parameters associated with the highest validation BLEU score. We implement a greedy decoding mechanism by taking the argmax of the probability distribution over tokens at each step. We leave for future work other decoding mechanisms such as beam search (Graves, 2012).

## 6 EXPERIMENTAL RESULTS

### 6.1 QUANTITATIVE EVALUATION OF GENERATED REPORTS

We begin by quantitatively evaluating the ability of the pre-training methods to generate high quality multilingual reports. In Table 1, we present the $BLEU-1$, METEOR, and $ROUGE-L$ scores of reports generated in seven different languages.

We find that RTLP performs on par with state-of-the-art generative pre-training methods. For example, on average, RTLP achieves $BLEU-1 = 28.5$ whereas MLM and MARGE achieve $BLEU-1 = 29.4$ and 28.9, respectively. This finding holds across languages and evaluation metrics. Furthermore, we find that RTLP outperforms the state-of-the-art *discriminative* pre-training method, ELECTRA. For example, on average, RTLP and ELECTRA achieve $ROUGE-L = 33.4$ and 0.5, respectively. We note that we were unable to achieve satisfactory performance with ELECTRA despite extensive experimentation (please see code for reproducibility). One hypothesis for this stems from the difficulty of optimizing its objective function which comprises a generative *and* discriminative term. We also find that, regardless of the pre-training method implemented, performance varies significantly across languages. For example, RTLP achieves $BLEU-1 = 19.8$ and 33.5 on Greek (el) and English (en) reports, respectively. We hypothesize that this is due to a high level of dissimilarity between the Greek vocabulary and that of the remaining languages. This, in turn, reduces the amount of knowledge transferred from the other languages to the Greek language. Overall, our findings suggest that RTLP can be an effective discriminative multilingual pre-training method. Qualitative evidence to support this claim is provided in the next section.

Table 1: **Multilingual cardiac signal captioning performance of pre-training methods.** Results are shown on the test set across five seeds. The standard deviation is shown in brackets. For clarity, we have highlighted our method in gray. We find that RTLP performs on par with state-of-the-art language pre-training methods, MLM and MARGE.

| Language Pre-training Method | German (de) | Greek (el) | English (en) | Spanish (es) | French (fr) | Italian (it) | Portuguese (pt) | Average |
|---|---|---|---|---|---|---|---|---|
| *BLEU-1* | | | | | | | | |
| MLM (Devlin et al., 2019) | 25.9 (0.6) | 20.5 (0.3) | 31.3 (0.5) | 33.2 (0.8) | 29.7 (0.6) | 30.3 (0.2) | 34.9 (0.7) | 29.4 (4.6) |
| ELECTRA (Clark et al., 2020) | 0.1 (0.1) | 0.2 | 0.2 (0.2) | 0.3 (0.1) | 0.6 (0.1) | 0.5 (0.1) | 0.5 (0.1) | 0.3 (0.2) |
| MARGE (Lewis et al., 2020) | 24.9 (1.0) | 19.5 (0.9) | 30.8 (0.5) | 32.9 (0.5) | 29.7 (0.6) | 29.4 (0.5) | 34.5 (1.0) | 28.9 (4.8) |
| RTLP | 25.4 (1.1) | 19.8 (0.6) | 30.0 (0.7) | 33.1 (0.9) | 28.3 (0.8) | 30.0 (0.1) | 33.5 (1.0) | 28.5 (4.5) |
| *METEOR* | | | | | | | | |
| MLM (Devlin et al., 2019) | 36.7 (1.0) | 23.6 (0.2) | 37.3 (1.1) | 38.6 (0.6) | 33.5 (0.7) | 33.9 (0.7) | 38.8 (0.7) | 34.6 (5.0) |
| ELECTRA (Clark et al., 2020) | 0.3 (0.5) | 0.2 (0.1) | 0.2 (0.1) | 0.5 (0.4) | 1.1 (0.3) | 0.9 (0.2) | 0.5 (0.2) | 0.5 (0.4) |
| MARGE (Lewis et al., 2020) | 35.6 (1.5) | 22.2 (1.0) | 36.5 (0.8) | 37.1 (0.6) | 33.1 (1.0) | 32.9 (0.9) | 37.8 (1.1) | 33.6 (5.1) |
| RTLP | 36.5 (0.7) | 22.6 (1.0) | 36.0 (0.9) | 38.5 (0.8) | 32.4 (0.4) | 33.7 (0.9) | 37.6 (0.6) | 33.9 (5.1) |
| *ROUGE-L* | | | | | | | | |
| MLM (Devlin et al., 2019) | 34.6 (0.8) | 11.4 (1.9) | 28.5 (0.2) | 39.3 (1.3) | 34.5 (0.8) | 36.9 (0.5) | 39.1 (0.6) | 33.5 (9.3) |
| ELECTRA (Clark et al., 2020) | 0.2 (0.3) | 0 | 0.2 | 0.5 (0.3) | 1.0 (0.2) | 0.8 (0.1) | 0.5 (0.1) | 0.5 (0.4) |
| MARGE (Lewis et al., 2020) | 33.2 (0.7) | 11.1 (2.3) | 38.1 (0.5) | 39.2 (0.6) | 34.4 (0.8) | 36.1 (0.6) | 39.0 (0.5) | 33.0 (9.4) |
| RTLP | 34.0 (1.1) | 11.6 (2.3) | 36.3 (0.9) | 39.1 (1.2) | 33.1 (0.9) | 36.5 (1.0) | 37.3 (0.9) | 32.6 (8.9) |

### 6.2 QUALITATIVE EVALUATION OF GENERATED REPORTS

We also manually inspect the multilingual clinical reports generated by RTLP. Ideally, such reports should accurately reflect clinical information. In Fig 2, we present a 12-lead ECG segment alongside the multilingual ground-truth reports and those generated by RTLP and MLM.

There are three main takeaways from Fig 2. First, RTLP allows networks to generate reports that accurately capture the aberrant morphology (shape) of the ECG signal. For example, in Spanish (es), the ground-truth and RTLP-generated reports both explicitly mention the cardiac abnormality *"bloqueo de rama izquierda"* (left bundle branch block). Second, we find that RTLP-generated reports manage to capture critical aspects of the ECG signal that their MLM counterparts struggle with. For example, in English (en), both the ground-truth and RTLP-generated reports both explicitly mention *"left hypertrophy possible"*, whereas this phrase is noticeably absent from the MLM-generated report. Such an absence is problematic as it might result in a physician overlooking this aspect, and thus failing to act accordingly. Lastly, we find that MLM-generated reports can include additional *erroneous* phrases which are neither present in the ground-truth report nor in the RTLP-generated report. For example, in Portuguese (pt), the MLM-generated report mentions *"atrial flutter"*, which is noticeably (and correctly) absent from the remaining reports. Similar erroneous inclusions can also be found in the German (de) and French (fr) reports. This is also problematic since physicians can be misled by such statements, potentially resulting in unnecessary medical treatments. These findings indicate that RTLP allows networks to generate reliable and clinically accurate reports.

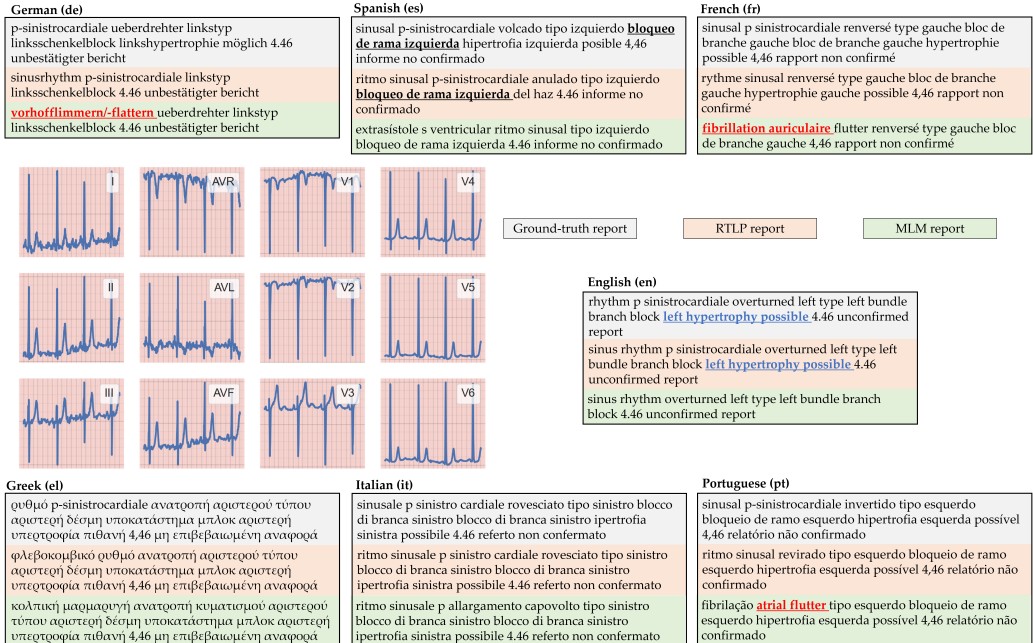

Figure 2: **12-lead ECG, multilingual ground-truth reports, and those generated by RTLP and MLM.** We show some phrases in **bold** which exhibit a high level of agreement in both the ground-truth report and that generated by RTLP, in blue which are captured by RTLP and *not* captured by MLM (false negatives), and in red which MLM erroneously includes (false positives). Overall, we show that RTLP can generate reports that accurately capture the pathology of the cardiac signal.

## 6.3 QUANTIFICATION OF CLINICAL UTILITY OF GENERATED REPORTS

So far, we have qualitatively shown that RTLP is capable of generating clinically accurate and plausible reports. It could be argued, however, that such a qualitative evaluation presents a limited view of the clinical utility of these reports. More broadly, our evaluation setup can be questioned since we are comparing generated reports to potentially noisy multilingual "ground-truth" reports, which were, for the most part, generated via translation. In this section, we aim to allay this concern and demonstrate the clinical utility of our generated reports in a systematic manner, as explained next.

To demonstrate that the translated ground-truth reports exhibit clinically useful information, we examine their ability to identify characteristics (e.g., cardiac abnormalities) of the corresponding cardiac signal. The intuition is that a report which is highly predictive of such characteristics, which inform clinical decision-making, is likely to be of clinical utility. For example, this report could be

used for decision support. As such, we set out to learn a model that maps translated ground-truth reports to a single cardiac abnormality label (5 classes) associated with the ECG. We represent these reports as a bag-of-words (BoW) on which a Random Forest model is trained (see Appendix D for further details). In Fig. 3, we illustrate the performance of such models when trained and evaluated on mutually-exclusive ground-truth reports in different languages (**Target**). We find that ground-truth reports are indeed highly predictive of cardiac abnormalities. This is evident by the strong performance of these models (e.g., $AUC > 0.90$). Such a finding suggests that the translated ground-truth reports are of clinical value.

To demonstrate the clinical utility of the *generated* reports, we follow a procedure similar to the one just described with one exception. Instead of training a model on ground-truth reports, we do so on held-out reports that are generated by a framework (e.g., RTLP). Importantly, we evaluate all models on the same set of *ground-truth* reports. The intuition here is that high quality *generated* reports should allow for the learning of a predictive model that can generalize to the ground-truth reports. In Fig. 3, we illustrate the performance of such models when trained on MLM- and RTLP-generated reports. There are two main findings here. First, we see that the generated reports exhibit lower clinical utility than ground-truth reports. This is evident by the $\downarrow AUC$ when comparing the **Target** setting to the **MLM** setting. For example, in German (de), $AUC \approx 0.92 \rightarrow 0.82$. Second, we show that the generated reports are predictive of cardiac abnormalities. This is evident by $\uparrow AUC$ of models trained on such reports. For example, in Italian (it), RTLP-generated reports lead to $AUC \approx 0.84$. Such a finding supports the claim that RTLP-generated reports are of clinical value.

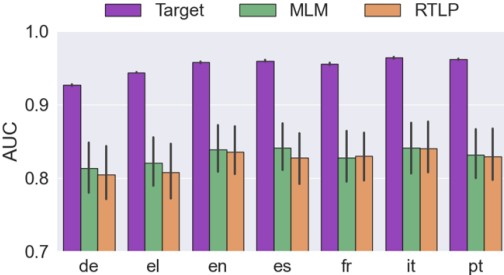

Figure 3: **Performance of models trained on clinical reports, either ground-truth or generated, to predict cardiac abnormalities.** In all experiments, a Random Forest model is trained either on ground-truth reports (**Target**) or those generated by MLM and RTLP. Models are evaluated on a mutually-exclusive set of ground-truth reports across five random seeds. Translated ground-truth reports and those generated, for example, by RTLP are predictive of cardiac abnormalities. Such a finding demonstrates the clinical utility of reports, and by extension, the system that generates them.

### 6.4 INVESTIGATION OF THE CURSE OF MULTILINGUALITY

Multilingual neural systems can experience the *curse of multilinguality* (Conneau et al., 2020b). Concisely, this attributes the potentially poorer performance of multilingual models relative to their monolingual counterparts to interference between the various languages. Intuitively, tasking a network with generating reports in multiple languages, analogous to multi-task learning (Caruana, 1993), can be too demanding and thus hinder its ability to generate sensible reports. We explore this curse through the lens of the quality and clinical utility of generated reports. To do so, we first pre-train our networks, as per usual, and fine-tune them in the *monolingual* setting. We then compare the quality (BLEU$-1$) and clinical utility (AUC) of the generated reports in the multilingual setting to those in the monolingual setting. Such a comparison is presented in Fig. 4.

In Fig. 4 (left column), we find that MARGE does not experience the curse of multilinguality when evaluated along the dimension of either performance or clinical utility. This can be seen by the similar performance achieved by MARGE irrespective of whether it is fine-tuned in the monolingual or multilingual setting. For example, when generating Spanish (es) reports, MARGE achieves BLEU$-1 \approx 32$ and $AUC \approx 0.80$ in both settings. Such a finding suggests that incorporating multiple languages into the fine-tuning process has little to no effect on the quality and clinical utility

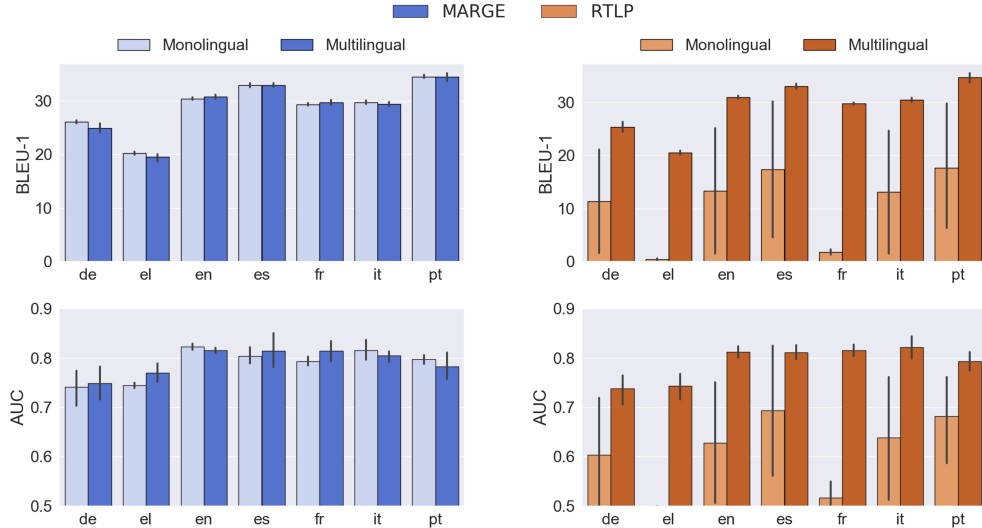

Figure 4: **Effect of multilinguality on the (top row) quality and (bottom row) clinical utility of the generated reports.** The multilingual setting involves simultaneously generating reports in all seven languages, $\mathbb{L} = \{\mathrm{de, el, en, es, fr, it, pt}\}$. Results are shown across five random seeds. Through the lens of report quality and clinical utility, MARGE does not consistently experience the curse of multilinguality. In contrast, RTLP benefits significantly from incorporating multiple languages into the fine-tuning process, a phenomenon we refer to as the *blessing of multilinguality*.

of the generated reports. This is in contrast to findings with RTLP (Fig. 4 right column), which reflect a significant benefit from the inclusion of multiple languages. This can be seen by the higher quality ($\uparrow$ BLEU$-1$) and clinical utility ($\uparrow$ AUC) of reports generated by RTLP in the multilingual setting than in the monolingual setting. For example, for French (fr) reports and in both settings, the network achieves BLEU$-1 \approx 30$ and $\approx 2$, respectively, and AUC $\approx 0.80$ and $0.52$, respectively. We denote this positive effect of multilinguality on the quality and clinical utility of reports as the *blessing of multilinguality*. We hypothesize that such benefits stem from the transfer and sharing of knowledge across languages. We also note the poorer performance of RTLP in the monolingual setting relative to the multilingual setting. For English (en) reports, RTLP achieves BLEU$-1 \approx 13$ and $\approx 31$, respectively. We hypothesize that this is due to the relative importance placed by the model on particular languages during multilingual RTLP pre-training. For example, such pre-training may implicitly weight languages differently. As such, the learned language-specific token representations may differ in their expressiveness. In light of this, networks fine-tuned in the monolingual setting may perform poorly.

## 7 DISCUSSION

In this paper, we proposed a neural multilingual cardiac signal captioning framework. In the process, we designed a discriminative multilingual pre-training method, RTLP, which randomly replaced tokens in a caption with those from a different language and tasked the network with identifying the language of all tokens. We showed that RTLP-generated reports exhibit high quality and clinical utility, and are on par with those generated by state-of-the-art pre-training methods such as MLM. We also showed that generated reports exhibit higher quality and clinical utility when RTLP is fine-tuned in a multilingual setting than in a monolingual setting, a phenomenon we refer to as the blessing of multilinguality. Based on these findings, we recommend exploiting RTLP for multilingual cardiac signal captioning. Several exciting paths remain. These include the design of a captioning framework that incorporates multiple modalities (e.g., medical videos, electronic health records). Doing so may result in higher quality reports and increase the likelihood of system adoption by medical professionals. In order to further validate our framework, we aim to exploit it to generate reports of a higher level of complexity.

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

# A DATASETS

## A.1 DATA PREPROCESSING

The ECG frames consisted of 2500 samples and consecutive frames had no overlap with one another. Data splits were always performed at the patient-level.

**PTB-XL** (Wagner et al., 2020). Each ECG recording was originally 10 seconds with a sampling rate of 500Hz. We extract 5-second non-overlapping segments of each recording generating frames of length 2500 samples. We follow the diagnostic class labelling setup suggested by Strodthoff et al. (2020) which resulted in five classes: Conduction Disturbance (CD), Hypertrophy (HYP), Myocardial Infarction (MI), Normal (NORM), and Ischemic ST-T Changes (STTC). Furthermore, we only consider ECG segments with one label assigned to them. The ECG frames were standardized to follow a standard Gaussian distribution.

## A.2 DATA SAMPLES

In this section, we outline the number of instances used during training.

Table 2: Number of instances (number of patients) used during training. These represent sample sizes for all 12 leads.

| Dataset | Train | Validation | Test |
|---------|-------|------------|------|
| PTB-XL | 22,670 (11,335) | 3,284 (1,642) | 3,304 (1,152) |

## A.3 VOCABULARY TOKENS

In this section, we outline the number of language-specific tokens available in each language's vocabulary for the two datasets.

Table 3: Number of language-specific tokens in each dataset

| Dataset | de | el | en | es | fr | it | pt |
|---------|------|------|------|------|------|------|------|
| PTB-XL | 2206 | 2662 | 1606 | 1950 | 1974 | 1866 | 2010 |

## B    TRANSLATION DETAILS

In this section, we outline the steps taken to translate the ECG reports originally found in the PTB-XL dataset. We remind readers that although these ECG reports are a mixture of English and German, they are predominantly in the latter. As a result, we treat German as the source language from which we translate the reports to other languages. More specifically, we follow these steps.

1. We leverage the Google Translate API[2] to first detect the source language of each ECG report. Although the majority of the reports are in German, some are in English, and this language detection step ensures that the ultimate translation is of a higher quality.

2. We continue to leverage the Google Translate API to translate ECG reports from the identified source language to the target language of interest.

3. Due to imperfections in the Google Translate API, certain ECG reports may not be translated in full or translated at all. To minimize the incidence of such cases, we repeat Step 2 several times and stop once we reach the following criterion: we deploy the language detection module of the Google Translate API on the translated reports to confirm that over 90% of them are indeed in the translated language. Although this implies that the final translated reports may have some noise, we found that this did not prevent our algorithm from learning appropriately.

---

[2]https://pypi.org/project/googletrans/

# C  IMPLEMENTATION DETAILS

## C.1  NETWORK ARCHITECTURES

In this section, we outline the neural network architectures used for our encoder and decoder. More specifically, we use the architecture shown in Table 4 for the encoder and that shown in Table 5 for the decoder.

Table 4: Encoder architecture used for experiments conducted on the PTB-XL dataset. $P$, $C_{in}$, and $C_{out}$ represent the kernel size, number of input channels, and number of output channels, respectively. A stride of 3 was used for all convolutional layers. $M$ represents the dimension of the final representation. We only use layer 5 when performing supervised pre-training. When captioning, layer 4 outputs $T$ temporal features.

| Layer Number | Layer Components | Kernel Dimension |
|:---:|:---:|:---:|
| 1 | Conv 1D
BatchNorm
ReLU
MaxPool(2)
Dropout(0.1) | $7 \times 12 \times 32$ ($P \times C_{in} \times C_{out}$) |
| 2 | Conv 1D
BatchNorm
ReLU
MaxPool(2)
Dropout(0.1) | $7 \times 32 \times 64$ |
| 3 | Conv 1D
BatchNorm
ReLU
MaxPool(2)
Dropout(0.1) | $7 \times 64 \times 128$ |
| 4 | Linear
ReLU | $128 \times M$ |
| 5 | Linear | $M \times$ C (classes) |

Table 5: Decoder architecture used for experiments conducted on the PTB-XL dataset. $E = 300$ represents the dimension of the representations from the encoder and the representations of the decoder tokens. $H = 4$ represents the number of heads used in each of the self and cross-attention modules. $C_{lang}$ represents the number of tokens in a specific language.

| Layer Number | Layer Components | Kernel Dimension |
|:---:|:---:|:---:|
| 1 | Transformer Decoder Layer | $E, H$ |
| 2 | Transformer Decoder Layer | $E, H$ |
| 3 | Transformer Decoder Layer | $E, H$ |
| 4 | Transformer Decoder Layer | $E, H$ |
| 5 | Linear | $E$ x $C_{lang}$ |

Table 6: Batchsize and learning rates used for training. The Adam optimizer was used for all experiments.

| Stage | Batchsize | Learning Rate |
|---|---|---|
| *Encoder* | | |
| Supervised Pre-training | 128 | $10^{-5}$ |
| *Decoder* | | |
| MLM Pre-training | 128 | $10^{-3}$ |
| ELECTRA Pre-training | 128 | $10^{-3}$ |
| RTLP Pre-training | 128 | $10^{-3}$ |
| MARGE Pre-training | 64 | $10^{-4}$ |
| *Combined* | | |
| Fine-tuning | 128 | $10^{-3}$ |

## C.2    ENCODER PRE-TRAINING

In this section, we outline the task used to pre-train the encoder of the captioning system in a supervised manner. Specifically, we learn an encoder, $f_\theta : u \in \mathbb{R}^{P \times D} \rightarrow y \in \mathbb{R}^C$ parameterized by $\theta$, that maps $P = 12$ $D$-dimensional ECG signals, $u$, (where $P$ represents the number of leads) to a $C$-dimensional output representing the probability assigned to each of the cardiac arrhythmia classes. When leveraging the PTB-XL dataset, $C = 5$. For a mini-batch of size, $B$, and where $c_i$ represents the ground-truth class for a particular instance, $x_i$, we learn this behaviour by optimizing the following categorical cross-entropy loss.

$$\mathcal{L}_{\mathcal{CE}} = -\frac{1}{B} \sum_{i=1}^{B} \log p_\theta(y_i = c_i) \tag{4}$$

We checkpoint, and eventually exploit, the parameters, $\theta$, that coincide with the lowest loss observed on the validation set. This ensures that we use parameters that do not exhibit overfitting.

## C.3 Baseline Implementations

**Masked Language Modelling.** Masked language modelling (MLM) can be thought of as analogous to a denoising autoencoder. Inputs are perturbed and the network is tasked with generating the original, unperturbed version of the input. In the context of natural language processing, a fraction $F = 0.15$ of the tokens in a sentence are chosen to be masked. Of these chosen tokens, $80\%$ are replaced with the token $[\text{MASK}]$, $10\%$ are replaced with a random token from the vocabulary, and the final $10\%$ are not replaced at all. The motivation behind this task lies in the ability of the network to leverage the context of masked tokens to correctly predict them. This, in turn, allows for the learning of rich representations. In our context, and to allow for a fair comparison to the multilingual pre-training methods, we follow the original implementation introduced by Devlin et al. (2019) for each of the language mini-batches. More specifically, at each iteration, we load $N$ mini-batches corresponding to $N$ languages and perform MLM on each of these batches.

**ELECTRA.** ELECTRA, as opposed to MLM introduced above, is a discriminative language representation learning method. ELECTRA builds upon the implementation of MLM in the following ways. First, instead of masking tokens and tasking the network with generating the original token, ELECTRA performs a binary classification of whether a token was replaced or not. The motivation for doing so lies in the alleged unnecessary complexity associated with generative language representation learning methods. Moreover, instead of replacing tokens with the $[\text{MASK}]$ token, ELECTRA proposes to do so by exploiting the predicted outputs of an MLM. This increases the likelihood that replaced tokens are in-distribution. As a result, ELECTRA simultaneously trains an MLM network and a binary classifier. In our context, we follow the original implementation introduced by Clark et al. (2020) for each of the $N$ mini-batches.

**MARGE.** MARGE is a generative multilingual language representation learning method that exploits source documents in various languages to generate text from a similar yet distinct target document. For example, $M$ source documents with $M * S$ tokens are encoded and leveraged by a decoder to generate the $T$ tokens in the target document. In doing so, the network is able to capture relationships between languages and thus learn representations useful for downstream multilingual tasks. In the original implementation (Lewis et al., 2020), similar documents need to be retrieved from a database. In our context, however, our ECG reports are available in $N$ different languages and thus the target document is formed by a report in one language and the source documents are formed by reports in the remaining $N - 1$ languages. Since our ECG reports were translated from a single original language, we used reports in this language as target documents. For PTB-XL, this amounts to using German.

## C.4 Evaluation Metrics

As we are mainly interested in the cardiac captioning task, we leverage three automatic metrics commonly used to evaluate image-captioning (BLEU score (Papineni et al., 2002), METEOR (Banerjee & Lavie, 2005) and ROUGE−L (Lin, 2004)). At a high-level, these metrics quantify the degree of overlap of n-grams between a ground-truth sentence and a generated sentence. An n-gram can be thought of as a combination of tokens (words) that neighbour one another. For example, a 1-gram simply consists of all individual tokens in a sentence whereas a 2-gram consists of all pairs of adjacent tokens in the sentence.

**BLEU Score.** The BLEU score first requires calculating the precision of n-grams for a particular value of n. This precision is defined as the number of overlapping n-grams between the generated and ground-truth sentence, $Overlap_n$, divided by the total number of n-grams in the generated sentence $Total_n$. Such a calculation is repeated for multiple values of $n \in [1, \ldots, N]$ before being averaged and weighted according to a brevity penalty, $BP$, which penalizes generated sentences which are shorter than the ground-truth sentence.

$$\text{BLEU} - \text{N} = BP \cdot \left( \Pi_{n=1}^{N} Precision_n \right)^{\frac{1}{N}} \qquad Precision_n = \frac{Overlap_n}{Total_n} \qquad (5)$$

**METEOR Score.** The METEOR score was designed, for the most part, to explicitly account for recall in n-gram overlap calculations, an operation not included in the BLEU score. Specifically, it aligns the ground-truth and generated sentences with one another, calculates the *unigram* (1-

gram) precision and recall, and derives an F-score with greater emphasis on recall than on precision. Similar to BLEU, it weights the F-score with a sentence brevity penalty, $BP$.

$$\text{METEOR} = \frac{10 \cdot Precision \cdot Recall}{9 \cdot Precision + Recall} \tag{6}$$

**ROUGE Score.** Although there exist multiple ROUGE scores (e.g., ROUGE-N, ROUGE-L, ROUGE-W, and ROUGE-S), we opt for the ROUGE-L score because it obviates the need to set a value of n for the n-grams. Formally, ROUGE-L calculates the F-score (geometric mean of precision and recall) of the longest common sub-sequence (LCS) of tokens between the ground-truth sentence with $m$ tokens and the generated sentence with $n$ tokens.

$$\text{ROUGE} - \text{L} = \frac{2 \cdot Precision \cdot Recall}{Precision + Recall} \tag{7}$$

$$Precision = \frac{LCS}{n} \quad \text{and} \quad Recall = \frac{LCS}{m}$$

## C.5  CLINICAL REPORT TOKENIZERS

Throughout the manuscript, we exploit SpaCy as the main natural language processing package. Specifically, we use language-specific pipelines that are publicly-available such as the following for the Italian language: `https://spacy.io/models/it#it_core_news_md`. Pipelines similar to this allow us to tokenize the reports in specific languages. For all languages, except for English, we use the pipelines with the suffix "core news md". For English reports, we use the pipeline with the suffix "core web md".

## D  QUANTIFICATION OF CLINICAL UTILITY OF REPORTS

In this section, we outline the implementation details for conducting the experiments that attempt to demonstrate the clinical utility of reports. In order to allay concerns about the quality of the translated ground-truth reports, and to demonstrate the clinical utility of generated reports, we propose to train a model that maps such reports to single-label cardiac abnormalities reflected in the corresponding cardiac signals. The intuition is that reports which are highly predictive of such abnormalities (which are used for clinical decision-making) are likely to be of clinical value.

From a mechanistic perspective, we consider the translated ground-truth reports found exclusively in a held-out set (3284 reports). We split this subset of reports into a training, validation, and test set using a $50 : 20 : 30$ split. This amounts to $1642$, $656$, and $986$ reports, respectively. Given the relatively small size of the corpus, and the simplicity of the reports, and to avoid overfitting, we opted for a straightforward bag-of-words (BoW) unigram feature representation (maximum features $=$ $50$). We train a Random Forest model on such representations in the training set to perform a multi-class classification of cardiac abnormalities. These abnormalities are Conduction Disturbance (CD), Hypertrophy (HYP), Myocardial Infarction (MI), Normal (NORM), and Ischemic ST-T Changes (STTC). We then evaluate the model on the mutually-exclusive set of reports in the test set.

In order to demonstrate the clinical utility of *generated* reports, we follow the same procedure as the one described above, with one exception. Instead of training on *ground-truth* reports, we now train on *generated* reports. In all cases, we continue to evaluate on the same set of *ground-truth* reports in order to allow for a fair comparison between the models in both settings. The intuition of this setup is, once again, that reports of higher clinical utility should allow for a model that is more predictive of cardiac abnormalities.

