# OpenReview forum: "Let Your Heart Speak in its Mother Tongue: Multilingual Captioning of Cardiac Signals"
_ICLR.cc/2022/Conference — ICLR 2022 Submitted_

### Official Review · Reviewer_VEwE · 2021-11-01

**Correctness:** 3
**Technical Novelty And Significance:** 3
**Empirical Novelty And Significance:** 3
**Recommendation:** 6
**Confidence:** 4

**Main Review:**

This paper proposed a novel multilingual pretraining language model settings. The idea is very interesting and the experiment setting is solid.   This paper conducted many result analyses which are helpful to understand the model's performance.

1. Using BLUE and ROUGE as the metrics is not enough in the evaluation of clinical report generation. The precision of content is critical in clinical reports. High BLUE/ROUGE does not represent the report is well written. In my opinion,  the performance of clinical text generation should be validated based on its content, although it is very hard and requires extensive manual work.

2. The gold-standard reports for non-English languages came from Google translation. This setting may lead to evaluation bias. It would be better to conduct a sensitivity analysis by using other translation tools.

3. It is not clear which network structure was used to represent the cardiac signals? Convolutional NN? Transformor? Other time series models?

4. It is not clear how to build the categorical distribution when selecting the semantic-similar target language tokens (page 4).

5. The "blessing of multilinguality"  comes from the low performance of RTLP in a single language. Based on this low baseline, the improvement from multilanguage has limited value. (RTLP + multilanguage can not beat MARGE).

**Summary Of The Paper:**

This paper proposed a new report generation system for cardiac signals. It applied replaced token language prediction (RTLP) settings to improve the report generation performance. Experiments show that the proposed RTLP framework can achieve comparable performance with SOTA models. Extensive analyses were conducted to examine the diversity and the performance in multilanguage settings.

**Summary Of The Review:**

The proposed model is novel and interesting. While the evaluation metrics in clinical text generation are not persuasive. And some of the important details were missing in the current draft.

---

> ### Author Response · Authors · 2021-11-12
> **Response to Reviewer VEwE - Round 1 (Part 1)**
>
> We would like to thank the reviewer for taking the time and effort to review our manuscript and for providing us with valuable feedback. We have addressed your comments below, modified the manuscript accordingly, and highlighted those changes in yellow.
>
> **Evaluating Clinical Report Generation**
>
> Designing suitable automated evaluation metrics (BLEU, ROUGE, METEOR) is an open research question within the field of natural language processing. Due to the distinct imperfections of these metrics, we opted to evaluate our framework on multiple metrics in order to obtain a holistic overview of its performance. As the reviewer pointed out, the precision of clinical reports is of utmost importance. However, performing a manual validation of generated reports can be non-trivial, as the reviewer points out. This challenge is especially pronounced in our context due to the requirement of a panel with cardiologists capable of communicating in multiple languages. We leave such manual validation to future work.
>
> **We do, however, conduct new experiments (Sec. 6.3 and 6.4) to demonstrate the clinical utility of the generated reports in a more systematic manner**. In short, we evaluate the ability of generated reports to predict cardiac abnormalities present in the corresponding cardiac signals. We refer the reviewer to Sec. 6.3 and 6.4 for an in-depth explanation of the experiments and the results. We show that RTLP-generated reports are predictive of cardiac abnormalities and thus exhibit high clinical utility (for example, reports can be used for decision support).
>
> **Network Architectures**
>
> We used a 1D convolutional network to extract features from the input cardiac signals. The details of this network are provided in Appendix C.1 (page 15). This relatively simple network was chosen based on previous publications that demonstrated its strong performance in discriminating between cardiac arrhythmias and our own empirical results (not shown) for discriminating between abnormalities in the PTB-XL dataset. We make reference to Appendix C.1 in Sec. 5 (paragraph 3, page 5).
>
> **Forming Categorical Distribution**
>
> The categorical distribution involves performing the softmax operation on the pairwise similarity between the embedding of the source token and the embeddings of all tokens in the target language. This, in turn, generates a probability mass function where the number of states is equal to the number of tokens in the target language. We can then sample from this distribution to select the target language token.
>
> **Blessing of Multilinguality**
>
> In Sec. 6.4 (end of paragraph 2, page 9), we described this phenomenon in depth and presented a hypothesis for the relatively poorer performance of RTLP in the monolingual setting. In Table 1, we compared the absolute performance of the pre-training frameworks. In this section, we transition to exploring the effect of multilinguality on the various pre-training frameworks. The main take-away message here is that multilinguality offers greater benefits to the RTLP pre-training framework than to MARGE. These benefits manifest in the form of improved quality and clinical utility of reports.

---

### Official Review · Reviewer_HBxE · 2021-11-02

**Correctness:** 3
**Technical Novelty And Significance:** 4
**Empirical Novelty And Significance:** 4
**Recommendation:** 8
**Confidence:** 4

**Main Review:**

Strengths:
1. This paper is clearly written. The paper is easy to follow and understand.
2. The targeted problems, i.e., cardiac signal captioning and multilingual captioning, are novel and important in both artificial intelligence and clinical medicine.
3. The proposed multilingual cardiac signal captioning system is well-motivated, novel, and interesting.
4. The experiments and analysis are extensive and solid.

Weaknesses:
1. The presentation can be further improved. In the Abstract and Introduction, can you give more explanations about "Generating these reports, however, can be time-consuming and error-prone, while also exhibiting a high degree of intra- and inter-physician variability"? For example, why generating these reports can be error-prone? What problems will be brought by the high degree of intra- and inter-physician variability (can you give some examples in the Introduction)?

2. Although the targeted problems are important and novel, after reading this paper, I am still confused about how your proposed method is related to this cardiac signal captioning. I think the proposed approach can be used for conventional image captioning as well. In other words, have you solved the challenges and problems that are unique to the multilingual cardiac signal captioning?

3. The experiment should be improved. Firstly, the evaluation metrics used in this paper, e.g.,  BLEU and ROUGE, are all general metrics for text generation tasks. So, it is unclear why your proposed approach can bring improvements. In what aspects can the proposed method improve the performance of the multilingual cardiac signal captioning? Secondly, in Table 1, why is your proposed method lower than the baselines in some settings? Can you give more explanations? Thirdly, the Google Translate model is not specifically designed for biomedical texts, so you can give more analysis of the Google Translate model.

4. The paper is written in an optimistic tone that leads the reader to assume the proposed approach is rather good. However, I am more interested in knowing if the approach brings errors? And what type of errors does it bring? And why?

5. The related work is insufficient. It is suggested to add more discussions about the report generation for other types of medical signals, e.g., chest X-rays, which have been widely explored in existing papers [1][2][3][4][5][6][7].

Missing References:

[1] On the Automatic Generation of Medical Imaging Reports. In ACL, 2018

[2] Show, Describe and Conclude: On Exploiting the Structure Information of Chest X-ray Reports. In ACL, 2019.

[3] When Radiology Report Generation Meets Knowledge Graph. In AAAI, 2020.

[4] Exploring and Distilling Posterior and Prior Knowledge for Radiology Report Generation. In CVPR, 2021.

**Summary Of The Paper:**

This paper aims to build a multilingual cardiac signal captioning system to generate ECG reports, which describe the clinical findings in the input of electrocardiogram (ECG) signals. In particular, the proposed system can generate desirable and fluent reports in multiple languages, i.e., German, Greek, English, Spanish, French, Italian, and Portuguese. The experiments on a public dataset verify the effectiveness of the proposed approach, which performs on par with state-of-the-art language pre-training methods.

**Summary Of The Review:**

The paper is well-written and the motivation sounds reasonable. The targeted problems, i.e., cardiac signal captioning and multilingual captioning, are novel and important. The presented approach is novel and interesting. Thus, I tend to accept this paper.

---

> ### Author Response · Authors · 2021-11-12
> **Response to Reviewer HBxE - Round 1 (Part 1)**
>
> We would like to thank the reviewer for taking the time and effort to review our manuscript and for providing us with valuable feedback. We have addressed your comments below, modified the manuscript accordingly, and highlighted those changes in yellow.
>
> **Clarity**
>
> In the introduction, we provided the clinical motivation for designing a cardiac signal captioning framework. The current approach to generating ECG reports can involve a cardiologist manually writing reports. However, the manual aspect of this process means that errors in the report can occur. For example, errors of omission could occur whereby the physician accidentally omits a medical detail from the report. As for intra- and inter-physician variability, this implies that reports of the same cardiac signal generated by distinct physicians may exhibit differences. This could be due to different cardiologist expertise levels, training backgrounds, and preferences, to name a few.
>
> **Extension to Image Captioning**
>
> Conventional image captioning typically involves doing so in a single language (e.g., English). The core contribution of our system is the development and evaluation of a multilingual cardiac signal captioning framework. To prepare a network for such multilinguality, we proposed a multilingual pre-training method, RTLP. From this perspective, the design of RTLP was motivated by the presence of clinical reports in multiple languages and the desire to learn rich representations of language-specific tokens. As it stands, we are the first to propose such an approach in the context of cardiac time-series signals. Moving forward, we envision our framework also having an impact on multilingual captioning systems for other modalities, such as medical images and videos (mentioned in Discussion section).
>
> **Evaluation**
>
> In Table 1, we hypothesize that the variable performance of RTLP across languages is due to the implicit importance that is placed on the distinct languages during the pre-training and fine-tuning process. Languages on which greater importance is placed are more likely to exhibit improved results. We provided an in-depth description of this hypothesis in Sec. 6.4 (end of paragraph 2, page 9).
>
> Indeed, ground-truth translations (via Google Translate API) of reports can exhibit noise. However, we believe the degree of this noise is limited for two reasons. First, conceptually the relative simplicity of these ECG reports (i.e., they are shorter than X-ray reports and exhibit less variability in text), implies that the degree of noise is unlikely to be great. Second, from an empirical perspective, we show in new experiments (Sec. 6.3) that the translated ground-truth reports are still highly-predictive of cardiac abnormalities reflected in cardiac signals. This implies that they continue to exhibit clinical value.
>
> **Related Work**
>
> We agree with the reviewer that medical image captioning is indeed a rich field. In the modified version of the manuscript, we have now included a subset of the suggested publications.

---

> > ### Comment · Reviewer_HBxE · 2021-11-19
> > **Additional questions**
> >
> > Thanks for your responses and additional results. I have some additional questions:
> >
> > 1. Like any model supporting clinical decision making, measuring and understanding the faithfulness of the model output is important. As the proposed cardiac signal captioning system is evaluated by its ability to generate fluent output, faithfulness can be a challenge to these models. Can you prove it?
> > 2. In your illustrated examples (Figure 2), how do you force the model to generate the number 4.46?
> > 3. The evaluation metrics used in this paper, e.g., BLEU and ROUGE, are all general metrics for text generation tasks. So can you conduct a brief human evaluation to prove the effectiveness of the approach compared with MLM?
> > 4. I noticed that the cardiac signal captioning systems have been explored in some existing literature, so I would like to know what is the advantages of multilingual in real-life clinical settings?

---

### Official Review · Reviewer_VuBG · 2021-11-04

**Correctness:** 2
**Technical Novelty And Significance:** 3
**Empirical Novelty And Significance:** 3
**Recommendation:** 3
**Confidence:** 4

**Main Review:**

Strengths: This paper is very well written. Multilingual text generation is an interesting task, and more multilingual work is needed in the ML for health space.

Weaknesses: I have several concerns about the clinical utility of this task as well as the evaluation approach.
-	First of all, I think clarification is needed to describe the utility of the task setup. Why is the task framed as generation of the ECG report rather than framing the task as multi-label classification or slot-filling, especially given the known faithfulness issues with text generation? There are some existing approaches for automatic ECG interpretation. How does this work fit into the existing approaches? A portion of the ECG reports from the PTB-XL dataset are actually automatically generated (See Data Acquisition under https://physionet.org/content/ptb-xl/1.0.1/). Do you filter out those notes during evaluation? How does your method compare to those automatically generated reports?
-	A major claim in the paper is that RTLP generates more clinically accurate reports than MLM, yet the only analysis in the paper related to this is a qualitative analysis of a single report. A more systematic analysis of the quality of generation would be useful to support the claim made in the appendix. Can you ask clinicians to evaluate the utility of the generated reports or evaluate clinical utility by using the generated reports to predict conditions identifiable from the ECG? I think that it’s fine that the RTLP method performs comparable to existing methods, but I am not sure from the current paper what the utility of using RTLP is.
-	More generally, I think that this paper is trying to do two things at once – present new methods for multilingual pretraining while also developing a method of ECG captioning. If the emphasis is on the former, then I would expect to see evaluation against other multilingual pretraining setups such as the Unicoder (Huang 2019a). If the core contribution is the latter, then clinical utility of the method as well as comparison to baselines for ECG captioning (or similar methods) is especially important.
-	I’m a bit confused as to why the diversity of the generated reports is emphasized during evaluation. While I agree that the generated reports should be faithful to the associated ECG, diversity may not actually be necessary metric to aim for in a medical context. For instance, if many of the reports are normal, you would want similar reports for each normal ECG (i.e. low diversity).
-	My understanding is that reports are generated in other languages using Google Translate. While this makes sense to generate multilingual reports for training, it seems a bit strange to then evaluate your model performance on these silver-standard noisy reports. Do you have a held out set of gold standard reports in different languages for evaluation (other than German)?

Other Comments:
-	Why do you only consider ECG segments with one label assigned to them? I would expect that the associated reports would be significantly easier than including all reports.
-	You might consider changing the terminology from “cardiac arrythmia” categories to something broader since hypertrophy (one of the categories) is not technically a cardiac arrythmia (although it can be detected via ECG & it does predispose you to them)
-	I think it’d be helpful to include an example of some of the tokens that are sampled during pretraining using your semantically similar strategy for selecting target tokens. How well does this work in languages that have very different syntactic structures compared to the source language?
-	Do you pretrain the cardiac signal representation learning model on the entire dataset or just the training set? If the entire set, how well does this generalize to setting where you don’t have the associated labels?
-	What kind of tokenization is used in the model? Which Spacy tokenizer?
-	It’d be helpful to reference the appendix when describing the setup in section 3/5 so that the reader knows that more detailed architecture information is there.
-	I’d be interested to know if other multilingual pretraining setups also struggle with Greek.
-	It’d be helpful to show the original ECG report with punctuation + make the ECG larger so that they are easier to read
-	Why do you think RTLP benefits from fine-tuning on multiple languages, but MARGE does not?



**Summary Of The Paper:**

The goal of this paper is to develop an approach for generating multilingual ECG reports. The authors propose a new multilingual pretraining method in which tokens are randomly replaced with those from a different language, and the model must learn to identify the language of all tokens. The method – RTLP – performs similarly to MLM, ELECTRA, and MARGE according to BLEU-1, METEOR, and ROUGE-L metrics on generation of ECG reports, and the authors qualitatively assess that the generated reports are clinically accurate. The paper compares monolingual and multilingual versions of the models and find that RTLP benefits from multilingual training.

**Summary Of The Review:**

Overall, the combined unclear clinical utility, lack of support for a major claim in the paper (that RTLP generates more clinically accurate reports), and concerns about the evaluation strategies lead me to recommend this paper for rejection.

---

> ### Author Response · Authors · 2021-11-12
> **Response to Reviewer VuBG - Round 1 (Part 1)**
>
> We would like to thank the reviewer for taking the time and effort to review our manuscript and for providing us with valuable feedback. We have addressed your comments below, modified the manuscript accordingly, and highlighted those changes in yellow.
>
> **Motivation Behind Task Formulation**
>
> In Section 1 (paragraph 1, page 1), we provide the clinical motivation for the utility of a system that is capable of generating ECG reports. In short, this motivation focuses on reducing the burden associated with manually generating such reports, standardizing their generation to mitigate potential errors, and improving healthcare accountability.
>
> As for framing the task as ECG report generation instead of one of multi-label classification, we believe the former can offer several advantages. As an example, ECG reports, particularly when generated by cardiologists, are likely to exhibit a more holistic picture of the cardiac state of the patient compared to the presence of multiple labels. Of course, the validity of this statement would depend on the nature of such labels (e.g., how exhaustive they are, degree of granularity, etc.). With the PTB-XL dataset, for example, a multi-label setup would have obscured details present in the ECG reports.
>
> As for issues related to the faithfulness of text generation systems, we agree that this is an open question that researchers in the field of natural language processing are attempting to address. However, to be able to begin addressing this question in the context of cardiac time-series signals *and* in the context of multilingual reports, there needs to be a starting point for such research. We believe that our work offers such a starting point, given that it is the first to propose a multilingual captioning framework for cardiac signals. Moreover, as illustrated in Fig. 2, we show that the reports that are generated by RTLP can, in some cases, be faithful to the ground-truth reports. Such a finding provides evidence that text generation systems do have the potential to generate accurate and clinically-plausible ECG reports.
>
> We have **modified the introduction to include the motivation** behind formulating a text generation system as opposed to a multi-label classification system (Sec. 1, paragraph 2, page 1)
>
> **Comparison to ECG Interpretation Systems**
>
> As the reviewer points out, there exists some work on, and even industry-grade systems for, automatic ECG interpretation. We view our work as supplementing these existing systems which have not addressed the task of multilingual captioning of cardiac time-series signals. For example, although we have not introduced this topic in the context of this paper, a potential downstream application of our system is one where multiple candidate ECG reports are recommended to a cardiologist who can then decide on a single report and add details if need be.
>
> **PTB-XL Data Pre-processing**
>
> With regards to the reports in the PTB-XL dataset, we do not filter out the reports which, as stated on their website, are automatically generated via an interpretation device. The main reason for not doing so is that the meta-data that indicates whether a report was generated by a cardiologist or automatically is not available. This, in turn, precludes our ability to filter reports accordingly. As such, during inference, it is likely that we are also evaluating the reports generated by our framework against reports which are automatically generated (via ECG interpretation device).
>
> **Quantifying Clinical Utility of Framework**
>
> We appreciate the reviewer's suggestion on more systematically evaluating the reports generated by the various frameworks. **We have addressed this by conducting an extensive set of new experiments which are now presented in Sec. 6.3 and 6.4**. In short, we evaluate the clinical utility of the generated reports by exploiting them to predict cardiac abnormalities reflected in the ECG. We show that RTLP-generated reports are of high clinical utility and exhibit similar clinical utility to reports generated by MLM. We are more than happy to receive additional feedback on these sections in order to improve the delivery of the content.

---

> > ### Author Response · Authors · 2021-11-12
> > **Response to Reviewer VuBG - Round 1 (Part 2)**
> >
> > **Contributions + Comparison to Previous Work**
> >
> > The single core contribution of our work is the development and evaluation of a multilingual cardiac signal captioning framework. To the best of our knowledge, we are the first to propose such a framework. We agree with the reviewer that demonstrating the clinical utility of our framework is critical. For this reason, we outlined the clinical motivation for and utility of this framework in Sec. 1 (paragraph 1, page 1). Our new experiments (Sec. 6.3 and 6.4) also demonstrate clinical utility more systematically and in an empirical manner.
> >
> > With regards to comparing to baselines in the domain of ECG captioning, that is non-trivial given that we are the first to propose a multilingual captioning framework for ECG signals. Moreover, other captioning frameworks in the realm of medical imaging are limited to a single language, and thus do not naturally extend to the multilingual setting.
> >
> > **Diversity of Generated Reports**
> >
> > We agree with the reviewer that diversity can be a misleading metric to rely upon in the context of medical text generation. For example, as the reviewer pointed out, if a clinical dataset happens to be biased towards a normal condition (i.e., severe class imbalance), then we might expect ground-truth reports to be more similar to one another. As such, low diversity reports may not necessarily be problematic.
> >
> > In light of the greater importance of demonstrating the clinical utility of the generated reports than examining the diversity of such reports, we have now replaced the content in the manuscript that used to refer to diversity with content that now refers to clinical utility. Specifically, these changes have affected Sec. 6.3 and 6.4. We believe this presents a more cogent argument for the overall utility of a multilingual cardiac signal captioning framework.
> >
> > **Quality of Translated Ground-Truth Reports**
> >
> > We are indeed using multilingual ECG reports for pre-training, fine-tuning, and evaluating the framework. Such ground-truth translations of reports can exhibit noise. However, we believe the degree of this noise is limited for two reasons. First, conceptually, the relative simplicity of these ECG reports (i.e., they are shorter than X-ray reports and exhibit less variability in text), implies that the degree of noise is unlikely to be great. Second, from an empirical perspective, we show in new experiments (Sec. 6.3) that the translated ground-truth reports are still highly-predictive of cardiac abnormalities reflected in cardiac signals. This implies that they continue to exhibit clinical value.
> >
> > Having said this, we do not currently have access to a gold standard set of reports in different languages. Curating such a set of reports would require a panel of expert cardiologists capable of communicating in different languages. Unfortunately, assembling such a panel is non-trivial and may also suffer from noise (e.g., due to inter-physician variability in translations).
> >
> > **PTB-XL Data Pre-processing**
> >
> > We only considered ECG reports associated with a cardiac signal corresponding to a single label for two reasons. First, as the reviewer pointed out, it simplifies the learning problem since the remaining reports may be less complex. Second, and more importantly, such multi-label signals are quite rare in the considered dataset (<2%). This insufficient coverage would make it difficult to meaningfully learn how to generate their corresponding reports. We leave it to future work to generate such reports (mentioned in Discussion section).
> >
> > **Terminology**
> >
> > We thank the reviewer for bringing this to our attention. In the camera-ready version of the manuscript, we will replace the term 'cardiac arrhythmia' with 'cardiac abnormality' to avoid any confusion.
> >
> > **Implementation Details**
> >
> > We pre-train the cardiac signal representation learning model exclusively on the training set (this is now mentioned in Sec. 5, paragraph 2). This allows for a more rigorous evaluation during inference on the test set since these cardiac signals are essentially unseen before. On a tangential note, it would be interesting to explore our framework in the context of out-of-distribution generalization (e.g., when cardiac signals reflect cardiac abnormalities unseen before) and even continual learning (e.g., generating reports to new cardiac signals streaming over time). We leave such explorations to future work.
> >
> > We use language-specific Spacy tokenizers for all of our experiments. For example, when tokenizing reports in Italian, we leverage the guidelines provided here (\url{https://spacy.io/models/it#it_core_news_md}). Similar guidelines exist for the remaining languages. A new Appendix C.5, which is refered in Sec. 5 (paragraph 3), now includes these implementation details.
> >
> > In Sec. 5 (paragraph 3), we have now made reference to Appendix C.1 which describes the network architectures.

---

> > > ### Author Response · Authors · 2021-11-12
> > > **Response to Reviewer VuBG - Round 1 (Part 3)**
> > >
> > > **Presentation**
> > >
> > > While presenting the original ECG report *with* punctuation would make it easier to read such reports, it might confuse readers when comparing them to non-punctuated reports generated by RTLP and other frameworks. We are happy to include these if the reviewer still believes this would be helpful. We have also expanded the size of the 12-lead ECG signals presented in Fig. 2.
> > >
> > > **Effect of Multilinguality on MARGE and RTLP**
> > >
> > > We addressed the benefit of multilingual fine-tuning relative to its monolingual counterpart in Sec. 6.4 (end of paragraph 2, page 9). We also provided a hypothesis for this relative improvement. In short, the outcome here depends on the framing of the results. MARGE performs as well in the multilingual setting as in the monolingual setting (please see Fig. 4). In contrast, RTLP performs better in the multilingual setting than in the monolingual setting. This is primarily due to the poor performance of RTLP in the monolingual setting, an in-depth description of which we provide in the same section of the manuscript.
> > >
> > > **Pre-training + Greek Language**
> > >
> > > From an empirical perspective, we have also shown that MARGE (a generative multilingual pre-training method) also struggles with generating reports in the Greek language (please see Table 1). More generally, however, recent research has demonstrated strong performance on the Greek language https://aclanthology.org/2020.emnlp-main.484.pdf. In contrast, in that paper, the authors struggle with other languages such as Swahili. We believe which language the system struggles with is multifactorial, dependent on the relative amount of data available in the language, the type of languages included in the multilingual training setup, the type of tokenization used, and so forth. These explorations, although interesting, are beyond the scope of this work.

---

> > > > ### Comment · Reviewer_VuBG · 2021-12-02
> > > > **Updated Review Post Author Response**
> > > >
> > > > I thank the authors for their thorough author response, especially the additional clinical utility experiments in sections 6.3 and 6.4. The highlighted text changes were especially helpful in evaluating what new experiments were run. I do think that the paper is significantly improved, but I think that it would beneficial from additional work and still recommend rejection.
> > > >
> > > > ### Re: PTB-XL Data Pre-processing
> > > > The authors state that they were unable to filter out reports which are automatically generated by an interpretation device because “the meta-data that indicates whether a report was generated by a cardiologist or automatically is not available. “ However, the ptbxl_database.csv file does have “initial_autogenerated_report” and “validated_by_human” columns, which would allow you to do just that. In your resubmission, I strongly recommend that you filter to only those reports that are validated by cardiologists. I do not see the utility of “evaluating the reports generated by our framework against reports which are automatically generated (via ECG interpretation device)” as you state that you do in your rebuttal.
> > > >
> > > > ### Re. Comparison to ECG Interpretation Systems
> > > > I am not sure I follow the authors’ claim in the rebuttal that their work would supplement existing ECG interpretation systems as this isn’t supported by the current framing of their paper. Furthermore, I don’t think that the fact that this system is multilingual necessarily prohibits comparison to monolingual baselines. The paper showed that the multilingual fine-tuning outperforms its monolingual counterpart on across all languages – how does it compare to existing monolingual approaches for automated ECG interpretation?
> > > >
> > > > ### Re. Quantifying Clinical Utility of Framework
> > > > I appreciate the authors’ effort to add clinical validation to the paper to address my concerns. According to their new results, there is no difference in AUC between the MLM and RTLP models. While I do not believe that better performance is strictly necessary, I would encourage the authors in their resubmission to articulate further the benefits of RTLP over existing work or how it can enable future work.

---

### Official Review · Reviewer_KZQE · 2021-11-07

**Correctness:** 3
**Technical Novelty And Significance:** 2
**Empirical Novelty And Significance:** 2
**Recommendation:** 3
**Confidence:** 5

**Main Review:**

The strengths of the paper:
1. The paper is well presented and the problem introduced in the paper is interesting as well as important.
2. The authors have tried multiple pre-training methods to train their decoder and use a relevant auxiliary task to supplement the performance of the proposed model.
3. The reports generated by the proposed framework seems to be quite coherent and similar to the gold standard captions.

The weaknesses of the paper:
1. The authors have mentioned that though there has been work around EEG captioning (Biswal et al 2019, 2020), the previous papers did not explore the multilingual captioning of the cardiac signals. But in the case of the proposed framework the gold-label captions for languages other than English are trained on captions generated via Google Translate API. So the ceiling of the decoder model is as good as the Translate API itself. If that is the case, wouldn't it be better to just generate the English caption and then translate them via the API. The cited paper (Conneau et al) uses such a strategy to generate for different languages to augment the dataset and generate new translations which the model might not have seen.
2. I do not understand the use case of generating all languages at once. The more suitable framework would be where the report is generated as per the requirement in which the source language is provided as an input to the decoder (<source_lg such as en, fr etc.><START>).
3. If MLM performs as good or better, in most cases, for generating the captions. What would justify the use-case of RTLP? It is an interesting approach but I do not see an added advantage of the proposed model.
4. The application is highly interesting but almost all parts of the proposed framework/models have been explored in a similar setting except for the multilingual aspect, which I am having a hard time understanding the use case of. In the proposed problem formulation, I would try to learn a really good cardiac signal captioning 'en' model and then either use the API itself or train a translator model. This would reduce the model parameters and make the whole framework more efficient.

**Summary Of The Paper:**

The authors of the paper have introduced a multilingual cardiac signal captioning framework. The authors have proposed a neural framework that generates captions for the cardiac signals in multiple languages simultaneously. The authors add an auxiliary task of identifying the language of some of the tokens to improve the performance of the decoder. The proposed framework achieves on par performance with state of the art pre-training methods.

**Summary Of The Review:**

The paper proposed a solution to an interesting problem but as mentioned in the weakness section above I could not comprehend clearly the use-case of the multilingual decoder. But I would be highly interested in discussing with the authors and other reviewers during the rebuttal period if I might be missing the aspect of using multilingual data for the framework.

---

> ### Author Response · Authors · 2021-11-12
> **Response to Reviewer KZQE - Round 1 (Part 1)**
>
> We would like to thank the reviewer for taking the time and effort to review our manuscript and for providing us with valuable feedback. We have addressed your comments below, modified the manuscript accordingly, and highlighted those changes in yellow.
>
> **Quality of Translated Ground-Truth Reports**
>
> Indeed, ground-truth translations of reports can exhibit noise. However, we believe the degree of this noise is limited for two reasons. First, conceptually, the relative simplicity of these ECG reports (i.e., they are shorter than X-ray reports and exhibit less variability in text), implies that the degree of noise is unlikely to be great (supported by empirical evidence). Second, from an empirical perspective, we show in new experiments (Sec. 6.3) that the translated ground-truth reports are still highly-predictive of cardiac abnormalities reflected in cardiac signals. This implies that they continue to exhibit clinical value.
>
> **Motivation for Multilinguality vs. Monolinguality + Translate**
>
> We believe that multilinguality confers multiple benefits to the overall learning process. We outlined these benefits in Sec. 4.1 (paragraph 1, page 3). As for the 'performance ceiling' of the decoder, even though the model is fine-tuned on noisy translations of the original report (multilingual setting), we believe that this 'ceiling' is not necessarily dictated by the quality of the translations generated by the Google Translate API. We empirically support this claim in Fig. 4 (right) where multilingual fine-tuning outperforms its monolingual counterpart on across all languages. Therefore, if we had only trained, for example, an English monolingual report generation system, we would have generated lower quality English reports. This, in turn, would have negatively affected any sort of translation conducted afterwards.
>
> **Motivation for Generating Reports in Multiple Languages**
>
> We agree with the reviewer that from the perspective of an end-user (e.g., cardiologist), our system would only be used to generate reports in a single language. This language is likely to be in the mother tongue of the cardiologists working within a hospital (e.g., French in France). To clarify, our framework *allows* for this use-case as we stated in Sec 4.1 (paragraph 1, page 3). Specifically, during inference, a user can decide to perform a forward pass while only exploiting the French-specific parameters of the decoder (in our case, that is simply a linear classification head). Other classification heads (which are associated with the remaining languages) need not be used.
>
> Although the suggestion that the reviewer puts forth (using a target language prefix token) is valid and typically used in neural machine translation (NMT), it would necessitate a common tokenization scheme (e.g., byte pair encoding) across languages. Instead, we opted to have distinct token dictionaries for each language. These distinct token dictionaries, in turn, allowed us to formulate the RTLP pre-training method (which depends on sampling tokens from different languages).
>
> **Number of Parameters / Efficiency**
>
> As for the number of parameters in our decoder, recall that the decoder consists of a backbone (i.e., transformer) that is *shared* across all languages. The only language-specific parameters are those contained in the linear classification heads. However, the number of such language-specific parameters is drastically less than the total number of parameters in the overall network. Moreover, the motivation for adopting this approach stems from recent findings which demonstrated the benefit of models with parameters that are both shared (backbone) and specific (linear classification heads) to languages. This is mentioned in Sec. 4.1 (paragraph 2, page 3).

---

> > ### Author Response · Authors · 2021-11-12
> > **Response to Reviewer KZQE - Round 1 (Part 2)**
> >
> > **Use-Case and Clinical Utility of RTLP**
> >
> > We believe that RTLP offers two distinct benefits relative to MLM (which was chosen as an exemplar *generative* language pre-training method).
> >
> > **First**, MLM was originally designed as a monolingual pre-training method and we adapted it to the multilingual setting. In contrast, RTLP, by design, optimizes a multilingual objective function during pre-training. This allows RTLP to learn representations of tokens from multiple languages, thus expanding its potential downstream applications (i.e., not limited to monolingual applications).
> >
> > **Second**, although the results in Table 1 illustrate that RTLP and MLM perform on par with another, such metrics, which are averaged across all generated reports, can conceal more subtle discrepancies in the generated reports. We outline a particular example of such discrepancies in Fig. 2 and demonstrate that MLM-generated reports can, in some cases, be less clinically plausible than those generated by RTLP. We understand that these qualitative evaluations can be limited. As such, we have also conducted new systematic evaluations, as explained next.
> >
> > We also conduct new experiments to more systematically determine the clinical utility of the reports that are generated by the various frameworks. **We refer the reviewer to the modified Sec. 6.3 and 6.4 for details of those experiments and results**. In short, we show that RTLP-generated reports are of clinical value *and* exhibit similar value to those generated by MLM.
> >
> > **Novelty**
> >
> > We would like to respectfully disagree with the reviewer's statement that the proposed framework has been explored in a similar setting. To clarify, we believe that the novelty of our overall framework is threefold. **First**, we propose a discriminative multilingual pre-training method (RTLP) whereas previous work has explored pre-training methods that are either generative and multilingual (e.g., MARGE) or discriminative and monolingual (e.g., ELECTRA). **Second**, to the best of our knowledge, we are the first to design a multilingual captioning framework for cardiac signals (previous work focused on medical images or EEG signals). **Third**, we conduct extensive experiments to demonstrate the quality and clinical utility of the generated reports and investigate the curse of multilinguality in the context of cardiac signals.

---

### Author Response · Authors · 2021-11-12
**Response to All Reviewers - High-Level Summary of Modifications**

We would like to thank the reviewers for taking the time and effort to review our manuscript and for providing us with valuable feedback. We have addressed your comments below, modified the manuscript accordingly, and highlighted those changes in yellow.

At a high-level, we have made the following changes, with more specific changes addressed in the reviewer-specific responses:

1. **Motivation** - we emphasized the motivation behind designing multilingual cardiac signal captioning systems.

2. **Claims** - we modified our claims in the abstract, introduction, and discussion sections to more accurately reflect the empirical results.

3. **Clinical Utility** - we conducted an extensive set of new experiments to demonstrate the quality and clinical utility of ground-truth and generated reports. (Sec. 6.3 and 6.4)

We look forward to an engaging discussion period!

---

### Decision · Program_Chairs · 2022-01-20

**Decision:**

Reject

**Comment:**

The paper addresses the problem of generating captions for ECG signals where they extend the task in the literature from monolingual to multilingual captions. The model proposed in this work is a variant of mask language model where they augment the target by switching the words from one language to another. In addition to predicting the actual words, they also predict the language associated of the words.

Pros
+ The problem is motivated by real world application and need.
+ The presentation is clear and the authors compare the performance of the model with appropriate recent models.

Cons
- The model has higher complexity in both training and parameters, yet achieves only comparable performance to simpler models.
- Empirical evaluation of the multi-lingual output is inadequate since the ​ground-truth is derived from Google Translate.
- There are also a number of other specific concerns raised by the reviewers (e.g., VuBG, HBxE).

Reviewers raised questions about several shortcomings. In response the authors updated the paper and were very engaged in the discussion with the reviewers. However, at the end of the day, the paper in its current form has serious shortcomings and left the reviewers unconvinced. I suggest the authors take advantage of the feedback from the reviewers and address them fully in their next iteration.